# Registered report: Wnt activity defines colon cancer stem cells and is regulated by the microenvironment

James Evans[1], Anthony Essex[1], Hong Xin[2], Nurith Amitai[2], Lindsey Brinton[3], Erin Griner[3], Reproducibility Project: Cancer Biology*

[1]PhenoVista Biosciences, San Diego, California; [2]Explora BioLabs, San Diego, California; [3]University of Virginia, Charlottesville, Virginia

*For correspondence:
tim@cos.io

Group author details
Reproducibility Project: Cancer Biology
See page 16

Competing interests:
See page 16

**Abstract** The Reproducibility Project: Cancer Biology seeks to address growing concerns about reproducibility in scientific research by replicating selected results from a substantial number of high-profile papers in the field of cancer biology. The papers, which were published between 2010 and 2012, were selected on the basis of citations and Altmetric scores (*Errington et al., 2014*). This Registered report describes the proposed replication plan of key experiments from 'Wnt activity defines colon cancer stem cells and is regulated by the microenvironment' by Vermeulen and colleagues, published in *Nature Cell Biology* in 2010 (*Vermeulen et al., 2010*). The key experiments that will be replicated are those reported in Figures 2F, 6D, and 7E. In these experiments, Vermeulen and colleagues utilize a reporter for Wnt activity and show that colon cancer cells with high levels of Wnt activity also express cancer stem cell markers (Figure 2F; *Vermeulen et al., 2010*). Additionally, treatment either with conditioned medium derived from myofibroblasts or with hepatocyte growth factor restored clonogenic potential in low Wnt activity colon cancer cells in vitro (Figure 6D; *Vermeulen et al., 2010*) and in vivo (Figure 7E; *Vermeulen et al., 2010*). The Reproducibility Project: Cancer Biology is a collaboration between the Center for Open Science and Science Exchange and the results of the replications will be published in *eLife*.

## Introduction

Wnt mediated activation of Frizzled receptors results in translocation of β-catenin to the nucleus where it binds to TCF/LEF transcription factors to induce expression of Wnt target genes. Wnt signaling proteins mediate a wide variety of biological processes during development, including maintenance of stem cell populations (*Malanchi and Huelsken, 2009*; *Clevers et al., 2014*), and aberrant Wnt activation is linked to several diseases, including cancer (*Clevers and Nusse, 2012*). Vermeulen and colleagues showed that high Wnt activity correlated with markers of colon cancer stem cells and enhanced clonogenic potential of cells (*Vermeulen et al., 2010*). They also showed that stromal myofibroblasts secreted factors such as HGF that enhanced Wnt activity and clonogenicity (*Vermeulen et al., 2010*). Further, treatment of more differentiated cells with myofibroblast conditioned medium (MFCM) enhanced Wnt activity in these cells and enhanced clonogenicity in vitro and in vivo, illustrating that colon cancer cell stemness can be modified by the microenvironment.

To assess Wnt activity in colon cancer cells, Vermeulen and colleagues utilized the TOP-GFP reporter system, a LEF-1/TCF responsive promoter driving expression of the enhanced GFP reporter (*Reya et al., 2003*). In Figure 2F, colon cancer stem cells were isolated from human colon cancer specimens and transduced with the TOP-GFP reporter. Wnt activity was then assessed using

fluorescence-activated cell sorting (FACS) in populations derived from a single cell, and flow cytometry was concurrently used to assess the levels of the cancer stem cell markers CD133, CD24/CD29 or CD44/CD166 in these cells. Vermeulen and colleagues found that cells with high Wnt activity levels correlated with expression of cancer stem cell markers (*Vermeulen et al., 2010*). Several other reports have used the TOP-GFP system or a similar reporter system to demonstrate that Wnt activity was enhanced in a population of cells expressing cancer stem cell markers. Correlation of cancer stem cells markers with high Wnt activity was found in primary and metastatic mouse mammary tumors (*Malanchi et al., 2012*), spheroid cultures of colon cancer cells (*Colak et al., 2014*), human and mouse colonic adenomas (*Prasetyanti et al., 2013*), and primary colon cancer cells (*Kemper et al., 2012*). All of these studies also found that cells with high Wnt activity maintained clonogenicity, while cells with low Wnt activity had markedly reduced clonogenic potential. In contrast, one report did find that while high Wnt reporter activity did correlate with expression of cancer stem cell markers, the tumorigenic capacity of cells was not dependent upon Wnt activity in four of the five cell lines tested, including three cell lines derived from primary colon cancers (*Horst et al., 2012*). The experiment presented in Figure 2F will be replicated in Protocol 2.

Figure 6D assessed the potential of MFCM or HGF to enhance the clonogenic potential in vitro of colon cancer cells with low Wnt activity (TOP-GFP$^{low}$ cells). Clonogenic potential was measured using a limiting dilution assay in which cells were plated at a range of densities and the number of colonies that grow over time was counted. This experiment also examined the ability of the small molecular c-Met inhibitor, PHA-665752, to block MFCM or HGF-mediated clonogenicity. Vermeulen and colleagues showed that treatment of TOP-GFP$^{low}$ cells with HGF or MFCM increased clonogenicity of these poorly clonogenic cells, and that clonogenicity could be reversed by c-Met inhibition (*Vermeulen et al., 2010*). Additionally, MET has been reported to be enriched in glioblastoma stem cells (GSCs) and promote their self-renewal (*Li et al., 2011*; *De Bacco et al., 2012*; *Joo et al., 2012*). Met inhibition could also reduce the clonogenic and tumorigenic potential of GSCs, with activation of the Wnt/β-catenin signaling pathway shown to be a key mediator of the HGF/Met signaling pathway in these cells (*Joo et al., 2012*; *Kim et al., 2013*). This experiment is replicated in Protocol 3. This result was expanded upon in Figure 7E to assess the effect of MFCM on TOP-GFP$^{low}$ cells in vivo, where Vermeulen and colleagues found that TOP-GFP$^{Low}$ cells coinjected with MFCM, or an admixture of the factors these cells secrete, had enhanced tumorigenicity compared to TOP-GFP$^{Low}$ cells injected alone (*Vermeulen et al., 2010*). To assess the affect of MFCM on tumorigenicity, limiting dilutions of the different cell populations, in the presence or absence of MFCM, were injected subcutaneously into nude mice and tumor formation was measured over time. Likewise, HGF, along with other cytokines secreted from tumor-associated cells, was demonstrated to increase the tumorigenic activity and metastatic potential of colorectal cancer progenitor cells (*Todaro et al., 2014*). This experiment is replicated in Protocol 4.

## Materials and methods

### Protocol 1: Isolation of colon cancer stem cells and infection with TOP-GFP

This experiment describes the isolation and culture of colon cancer stem cells. These spheroid cultures and the Co100 cell line used in the original study will be transduced with TOP-GFP and control plasmid. This will produce the single-cell-derived TOP-GFP clones used for further analysis in Protocols 2 and 3.

### Sampling

- This experiment will be performed once to generate two newly derived single spheroidal cultures, which along with the Co100 cell line will provide three colon cancer lines.
- Each of the three lines will be used to generate and isolate 3 different TOP-GFP cancer stem cell (CSC) clones and one control clone.

## Materials and reagents

| Reagent | Type | Manufacturer | Catalog # | Comments |
|---|---|---|---|---|
| Modified neurobasal A medium | Cell culture | Life Technologies | 10,888-022 | Original catalog # not specified |
| N2 supplement | Cell culture | Life Technologies | 17,502-048 | Original catalog # not specified |
| Lipid mixture-1 | Cell culture | Sigma–Aldrich | L0288 | Original catalog # not specified |
| Fibroblast growth factor—Basic, human (FGF) | Growth factor | Sigma–Aldrich | F0291 | Original brand not specified |
| Epidermal growth factor, human (EGF) | Growth factor | Sigma–Aldrich | E9644 | Original brand not specified |
| Human colon tissue fragments | Clinical sample | N/A | N/A | Original mainly used microsatellite stable primary tumors. |
| Phosphate buffered saline (PBS) without $MgCl_2$ and $CaCl_2$ | Buffer | Sigma–Aldrich | D8537 | Original brand not specified* |
| 100× Penicillin/streptomycin | Cell culture | Sigma–Aldrich | P4333 | Original brand not specified* |
| Amphotericin B | Cell culture | Sigma–Aldrich | A2942 | Replaces GIBCO brand used in original study* |
| DMEM/F12 medium with L-glutamine and sodium bicarbonate | Cell culture | Sigma–Aldrich | D8062 | Replaces GIBCO brand used in original study* |
| Collagenase, type 1A-S | Cell culture | Sigma–Aldrich | C9722 | Replaces Roche brand used in original study |
| Hyaluronidase from bovine testes, type 1-S, 400–1000 units/mg | Cell culture | Sigma–Aldrich | H3506 | Original brand not specified |
| 40 µm cell strainer | Labware | Corning | 431750 | Original brand not specified |
| Lympholyte-M | Chemical | Cedarlane | CL5031 | Original catalog # not specified |
| Co100 culture | Cell culture | Authors | N/A | From original lab |
| HEK293T cells | Cell line | ATCC | CRL-3216 | Included during communication with authors. Original brand not specified |
| Dulbecco's Modified Eagle's Medium (DMEM)—high glucose | Cell culture | Sigma–Aldrich | D6429 | Originally not specified |
| Fetal Bovine Serum (FBS) | Cell culture | Sigma–Aldrich | F0392 | Originally not specified |
| 0.5% trypsin/0.48 mM EDTA (5 mg/ml) | Cell culture | Sigma–Aldrich | T4174 | Included during communication with authors. Original brand not specified |
| 150 mm tissue culture plates | Labware | Corning | 430599 | Included during communication with authors. Original brand not specified |
| 100 mm tissue cultures plates | Labware | Corning | 430167 | Originally not specified |
| TOP-CMV-GFP reporter lentivirus vector | DNA construct | Authors | N/A | From original lab |
| psPAX2 packaging plasmid | DNA construct | Authors | N/A | From original lab |
| pMD2.G envelope plasmid | DNA construct | Authors | N/A | From original lab |
| *Trans*IT-293 | Transfection reagent | Mirus Bio | MIR 2704 | Originally not specified |
| 50 ml polypropylene conical tubes | Labware | Corning | 430290 | Included during communication with authors. Original brand not specified |
| OptiMEM-1 reduced serum medium | Cell culture | Life Technologies | 31985062 | Included during communication with authors. |
| Hexadimethrine bromide (polybrene) | Cell culture | Sigma–Aldrich | 107689 | Included during communication with authors. Original brand not specified |
| Propidium iodide | Chemical | Sigma–Aldrich | P4170 | Original brand not specified |
| FACS sorter | Equipment | BD Biosciences | FACSaria | |
| 96-well ultralow-attachment plate, flat bottom | Labware | Corning | 3474 | Original catalog # not specified |
| 24-well ultralow-attachment plate | Labware | Corning | 3473 | Included during communication with authors. |

* *Continued on next page*

*Continued*

| Reagent | Type | Manufacturer | Catalog # | Comments |
|---|---|---|---|---|
| 6-well ultralow-attachment plate | Labware | Corning | 3471 | Included during communication with authors. |
| 25 cm² ultralow-attachment tissue culture flask | Labware | Corning | 3815 | Original catalog # not specified |
| 75 cm² ultralow- attachment tissue culture flask | Labware | Corning | 3814 | Original catalog # not specified |

*From (**Todaro et al., 2007**).

## Procedure

Note:

- This protocol contains information described in *Todaro et al., 2007*.
- Fresh primary cells and Co100 cell line maintained in CSC medium: modified neurobasal A medium supplemented with 1× N2 supplement, lipid mixture-1 (1 ml/500 ml medium), fibroblast growth factor-basic (20 ng/ml), and epidermal growth factor (50 ng/ml) at 37°C in a humidified atmosphere at 5% $CO_2$.
- HEK293T cells maintained in DMEM supplemented with 10% FBS at 37°C with 5% $CO_2$.

1. Obtain twelve freshly excised human colon adenocarcinoma tissue fragments.
   a. If not enough viable spheroidal cultures are generated with the initial twelve fragments, an additional source of human colon adenocarcinoma cells will be obtained.
   b. Include histological diagnosis report and patient annotation.
   c. Note: with human colon tissue fragment samples, there is a 10–20% success rate of obtaining spheroidal cultures.
2. Wash 4 times in PBS supplemented with penicillin (500 U/ml), streptomycin (500 U/ml), and amphotericin B (1.25 µg/ml).
3. Incubate overnight in DMEM/F12 medium supplemented with penicillin (500 U/ml), streptomycin (500 U/ml), and amphotericin B (1.25 µg/ml).
4. Digest with collagenase (1.5 mg/ml) and hyaluronidase (20 µg/ml) in PBS for 1 hr at 37°C; shake repeatedly during digestion.
5. Pass the dissociated sample through a 40 µm cell strainer and wash with CSC medium.
6. Remove erythrocytes and cell debris by Lympholyte-M centrifugation following manufacturer's instructions.
7. Wash cells 2–3 times with CSC medium and maintain culture. Maintain Co100 cell line following same methodology.
   a. Once a viable culture is established, colon cancer cells will cluster into spheroids (∼50–100 cells/spheroid).
   b. Dissociate spheroids.
      i. Pellet spheroids by centrifuging 5 min at 1000 RPM.
      ii. Aspirate medium, being careful not to disrupt cell pellet.
      iii. Using a sterile 5 ml serological pipet, gently resuspend cells in 3 ml of 1 mg/ml trypsin by pipetting up and down 3×.
      iv. Place tubes in 37°C tissue culture incubator for 2.5 min.
      v. Agitate the cells by gently pipetting up and down 3 times using a sterile 5 ml serological pipet.
      vi. Return tubes to 37°C tissue culture incubator for an additional 2.5 min.
      vii. Stop the dissociation by adding 10 ml of DMEM supplemented with 10% FBS.
      viii. Pellet cells by centrifuging 5 min at 1000 RPM.
      ix. Aspirate medium, being careful not to disrupt cell pellet.
      x. Gently resuspend cells in an appropriate volume of CSC medium and perform a viable cell count.
   c. Cultures should be passaged when cell concentration exceeds $1 \times 10^6$ cells/ml of medium.
      i. Additionaly, maintain cultures, untransduced, as a control for Protocols 2, 3, and 4.
8. Transduce the two newly derived dissociated single spheroidal cultures and the Co100 cell line lentivirally with the TOP-GFP construct following the Trono lab Protocol for 'Production of Lentiviral Vectors in 293T cells' briefly described with the following modifications.
   a. Plate $4.7–5.8 \times 10^6$ HEK293T cells per 15 cm plate.

 b. Transfect 15 cm plate with TOP-GFP lentiviral vector, packaging plasmid, and envelope plasmid with *Trans*IT-293 transfection reagent following Trono lab protocol instructions.

 c. Change medium 6–8 hr later and add 15 ml/plate of fresh medium.

 d. Harvest 30 ml of medium per plate of transfected HEK293T cells (15 ml on day 1 and day 2 after transfection) and filter through a 0.45 µm filter.

 i. Store day 1 medium at 4°C until it can be combined with day 2 harvest.

 e. Concentrate by centrifugation (O/N at 4°C and 4000 RPM) in a 50 ml polypropylene conical tube.

 f. Resuspend virus pellet in 500 µl of Opti-MEM.

 g. Transduce dissociated spheroidal cultures with 20 µl of concentrated virus/$1 \times 10^6$ cells in 10 ml of medium supplemented with 8 µg/ml polybrene.

 h. Change medium after 24 hr of infection to remove polybrene, dead cells, etc.

9. After 3–4 passages and 2 weeks in culture, or when cells are growing robustly, sort for single, propidium iodide-negative, and GFP-positive cells by FACS.

 a. Dissociate spheroids as described in step 7b.

 b. Add 250 ng/ml propidium iodide solution directly to cells before analysis.

 c. Single cells were gated within the GFP positive population.

 i. Note: No specific level of GFP was used in the original study.

10. Deposit 1 cell/well in a 96-well ultralow-adhesion plate containing CSC medium.

 a. Plate at least three plates with single cells as the incidence of visible spheres is around 1%.

11. After visible spheres arise, transfer to ultralow-adhesion plates and expand three independent TOP-GFP clones from the two newly derived spheroidal cultures and the Co100 cell line.

 a. Once spheroid cultures arise from a single cell clone, transfer the spheroid into one well of a 48 well plate and gently break down the spheroid by mechanical dissociation.

 b. Slowly scale up to a larger culture surface (i.e., 1 well of a 48 well plate, then 4 wells of a 24 well plate, then 1 well of a 6 well plate, then 25 cm² flask, etc) by dissociating the culture (as described in step 7b) for the subsequent experiments (Protocols 2, 3, and 4).

12. Maintain clones from single-cell-derived TOP-GFP cells and untransduced parental spheroidal cultures, for further analysis (Protocols 2, 3, and 4).

## Deliverables

- Data to be collected:
  - Histological diagnosis and patient annotation of human colon tissue fragments.
  - All FACS plots in gating scheme (including controls), leading to final population of single, propidium iodide-negative, GFP-positive cells.
- Sample delivered for further analysis:
  - Spheroidal cultures (two newly derived and the Co100 cell line) as a control in Protocols 2, 3, and 4.
  - Single-cell-derived TOP-GFP clones (two newly derived and the Co100 cell line) for further analysis in Protocols 2, 3, and 4.

## Confirmatory analysis plan
N/A.

## Known differences from the original study
All known differences are listed in the materials and reagents section above with the originally used item listed in the comments section. All differences have the same capabilities as the original and are not expected to alter the experimental design.

## Provisions for quality control
A report of histological diagnosis and patient annotation will be included. Additionally, cell viability will be monitored during culture conditions. All of the raw data, including the FACS plots, will be uploaded to the project page on the OSF (https://osf.io/pgjhx) and made publically available.

## Protocol 2: flow cytometry analysis of CSC marker expression in TOP-GFP clones
This experiment will assess the association of TOP-GFP levels with CSC marker expression, specifically CD133, CD29, CD24, CD44, and CD166, which is a replication of the experiment reported in Figure 2F.

## Sampling

- This experiment will be performed with each of the 3 different TOP-GFP CSC clones from the two newly derived cultures and the Co100 cell line.
- Each TOP-GFP CSC clone will be analyzed for signal intensity for a total power of $\geq$ 80%.
  - See 'Power calculations' section for details.
- Staining conditions for each clone:
  - CD133.
  - CD24 and CD29.
  - CD24 alone.
  - CD29 alone.
  - CD44 and CD166.
  - CD44 alone.
  - CD166 alone.
  - Isotype controls.
  - Unstained control.
  - Untransduced spheroidal culture (no GFP) control.

## Materials and reagents

| Reagent | Type | Manufacturer | Catalog # | Comments |
|---|---|---|---|---|
| 0.05% trypsin/0.48 mM EDTA | Cell culture | Sigma–Aldrich | T3924 | Original brand not specified |
| Phosphate buffered saline (PBS) without MgCl$_2$ and CaCl$_2$ | Buffer | Sigma–Aldrich | D8537 | Originally not specified |
| Bovine serum albumin (BSA) | Chemical | Sigma–Aldrich | A3803 | Included during communication with authors. Original brand not specified |
| CD133 (clone AC133) -PE antibody (mouse IgG1) | Antibodies | Miltenyi Biotec | 130-098-826 | Use at 1:100 dilution. Conjugate selected by replicating lab. |
| CD44 (clone G44-26)—APC antibody (mouse IgG2b, κ) | Antibodies | BD Biosciences | 560890 | Use at 1:100 dilution. Conjugate selected by replicating lab. |
| CD166 (clone 105902)—PE antibody (mouse IgG1) | Antibodies | R&D Systems | FAB6561P | Original clone listed as 105901. Use at 1:100 dilution. Conjugate selected by replicating lab. |
| CD24 (clone ML5)—PE antibody (mouse IgG2a, κ) | Antibodies | BD Biosciences | 560991 | Use at 1:100 dilution. Conjugate selected by replicating lab. |
| CD29 (clone MAR4)—APC antibody (mouse IgG1, κ) | Antibodies | BD Biosciences | 561794 | Use at 1:100 dilution. Conjugate selected by replicating lab. |
| Anti-IgG1—PE, mouse (clone X-56) | Antibodies | Militenyi Biotec | 130-098-106 | Use at 1:100 dilution. Originally not specified. |
| Anti-IgG2b κ—APC, mouse (clone 27–35) | Antibodies | BD Biosciences | 555745 | Use at 1:100 dilution. Originally not specified. |
| Anti-IgG2a κ—PE, mouse (clone G155-178) | Antibodies | BD Biosciences | 555574 | Use at 1:100 dilution. Originally not specified. |
| Anti-IgG1 κ—APC, mouse (clone MOPC-21) | Antibodies | BD Biosciences | 555751 | Use at 1:100 dilution. Originally not specified. |
| Propidium iodide | Chemical | Sigma–Aldrich | P4170 | Original brand not specified |
| FACS sorter | Equipment | BD Biosciences | FACSaria | |

## Procedure
Note:

- TOP-GFP CSC clones, and untransduced spheroidal culture (no GFP) control, are generated in Protocol 1.

1. After obtaining single cell suspensions in Protocol 1, dissociate TOP-GFP CSC clones and untransduced cultures with trypsin as described in Protocol 1 and re-suspend $1 \times 10^6$ cells/ml in PBS supplemented with 1% BSA.

2. Stain cells with the following antibodies:
   a. CD133-PE (use at 1:100 dilution).
   b. CD24-PE (use at 1:100 dilution) and CD29-APC (use at 1:100 dilution).
   c. CD44-APC (use at 1:100 dilution) and CD166-PE (use at 1:100 dilution).
   d. Incubate antibodies with cells for 10 min in a dark refrigerator at 4°C (2–8°C is acceptable).
   e. Wash cells by adding 20× the reaction volume of PBS with 1% BSA and gently inverting tubes 3× (i.e., cell/antibody volume is 100 µl, add 2 ml PBS supplemented with 1% BSA). Centrifuge cells at 1000 RPM for 10 min. Carefully aspirate supernatant completely. Resuspend cells in 100 µl PBS.
   f. Include an unstained control for gating.
   g. Include untransduced spheroidal culture (no GFP) for gating.
   h. Include isotype control antibody stains.
      i. Anti-IgG1—PE.
      ii. Anti-IgG2b κ—APC.
      iii. Anti-IgG2a κ—PE.
      iv. Anti-IgG1 κ—APC.
3. Add 250 ng/ml propidium iodide solution to cells just before analysis.
   a. Include an unstained control for gating.
4. Perform flow cytometry analysis for the following populations:
   a. Analyze CD133 intensity:
      i. Gate for viable cells (propidium iodide-negative cells).
      ii. Gate for TOP-GFP expression.
         1. Identify top 10% and bottom 10% of TOP-GFP expression.
         2. Analyze CD133 intensity in at least 10,000 cells in each fraction.
      iii. Gate against a negative control (unstained cells).
   b. Analyze CD24 and CD29 intensity:
      i. Gate for viable cells (propidium iodide-negative cells).
      ii. Gate for TOP-GFP expression.
         1. Identify top 10% and bottom 10% of TOP-GFP expression.
         2. Analyze CD29 and CD24 intensity in at least 10,000 cells in each fraction.
      iii. Gate against a negative control (unstained cells) and cells stained with each antibody individually.
   c. Analyze CD44 and CD166 intensity:
      i. Gate for viable cells (propidium iodide-negative cells).
      ii. Gate for TOP-GFP expression.
         1. Identify top 10% and bottom 10% of TOP-GFP expression.
         2. Analyze CD44 and CD166 intensity in at least 10,000 cells in each fraction.
      iii. Gate against a negative control (unstained cells) and cells stained with each antibody individually.

## Deliverables

- Data to be collected:
  - All FACS plots in gating scheme (including controls), leading to final population of propidium iodide-negative, GFP-positive cells for analysis of each CD marker.
  - FACS mean fluorescence intensity and confidence intervals for each CD marker.

## Confirmatory analysis plan

This replication attempt will perform the following statistical analysis listed below and compute the effects sizes for each TOP-GFP CSC clone.

- Statistical Analysis:

  Note:

1. Since these tests will be performed for each of the three clones from the CSC cultures the alpha error will be adjusted with the Bonferroni correction.
   - Unpaired, two-tailed *t*-test with the Bonferroni correction for multiple comparisons:
     - CD133 expression from TOP-GFP^low cells compared to TOP-GFP^hgh cells.

- CD24 expression from TOP-GFP$^{low}$ cells compared to TOP-GFP$^{hgh}$ cells.
- CD29 expression from TOP-GFP$^{low}$ cells compared to TOP-GFP$^{hgh}$ cells.
- CD44 expression from TOP-GFP$^{low}$ cells compared to TOP-GFP$^{hgh}$ cells.
- CD166 expression from TOP-GFP$^{low}$ cells compared to TOP-GFP$^{hgh}$ cells.

- Meta-analysis of effect sizes:
  - Compare the effect sizes of the TOP-GFP CSC clones from the two newly derived cultures and the Co100 cell line (3 different clones each) and use a meta-analytic approach to combine the effects, which will be presented as a forest plot.

## Known differences from the original study

All known differences are listed in the materials and reagents section above with the originally used item listed in the comments section. All differences have the same capabilities as the original and are not expected to alter the experimental design.

## Provisions for quality control

Negative staining, individual antibody, and isotype controls are included to assess antibody staining relative to background. All of the raw data, including the FACS plots, will be uploaded to the project page on the OSF (https://osf.io/pgjhx) and made publically available.

## Protocol 3: clonogenicity assay of TOP-GFP CSC clones

This experiment will assess the effect of MFCM and recombinant HGF on the clonogenic potential of the TOP-GFP CSC clones. This experiment will also examine the ability of the small molecular c-Met inhibitor, PHA-665752, to block MFCM- or HGF-triggered clonogenicity. This is a replication of the experiment reported in Figure 6D.

## Sampling

- This experiment will be performed with each of the 3 different TOP-GFP CSC clones from the two newly derived cultures and the Co100 cell line and with each cohort assessing 96 wells for a total power of $\geq$ 80%.
  - See 'Power calculations' section for details.
- Each experiment has 8 cohorts:
  - Cohort 1: TOP-GFP$^{low}$ cells.
  - Cohort 2: TOP-GFP$^{high}$ cells.
  - Cohort 3: TOP-GFP$^{low}$ cells + HGF.
  - Cohort 4: TOP-GFP$^{low}$ cells + MFCM.
  - Cohort 5: TOP-GFP$^{low}$ cells + HGF + PHA-665752.
  - Cohort 6: TOP-GFP$^{low}$ cells + MFCM + PHA-665752.
  - Cohort 7: total TOP-GFP cells.
  - Cohort 8: total TOP-GFP cells + PHA-665752.
- Each cohort plates (per 96 well plate):
  - 1 cell $\times$ 24 wells.
  - 2 cells $\times$ 16 wells.
  - 4 cells $\times$ 8 wells.
  - 8 cells $\times$ 8 wells.
  - 16 cells $\times$ 8 wells.
  - 32 cells $\times$ 8 wells.
  - 64 cells $\times$ 8 wells.
  - 128 cells $\times$ 8 wells.
  - 256 cells $\times$ 8 wells.
    - The titration of cells might need to be adjusted depending on the clonogenic potential of the spheroid clones. An initial pilot experiment will be performed to assess the potential for the three populations (TOP-GFP$^{low}$, TOP-GFP$^{high}$, and total TOP-GFP), without treatment, before proceeding with this design.

## Materials and reagents

| Reagent | Type | Manufacturer | Catalog # | Comments |
|---|---|---|---|---|
| Modified neurobasal A medium | Cell culture | Life Technologies | 10,888-022 | Original catalog # not specified |
| N2 supplement | Cell culture | Life Technologies | 17,502-048 | Original catalog # not specified |
| Lipid mixture-1 | Cell culture | Sigma–Aldrich | L0288 | Original catalog # not specified |
| Fibroblast growth factor—Basic human | Cell culture | Sigma–Aldrich | F0291 | Original brand not specified |
| Epidermal growth factor human | Cell culture | Sigma–Aldrich | E9644 | Original brand not specified |
| Phosphate buffered saline (PBS) without $MgCl_2$ and $CaCl_2$ | Buffer | Sigma–Aldrich | D8537 | Original brand not specified |
| 0.05% trypsin/0.48 mM EDTA | Cell culture | Sigma–Aldrich | T3924 | Originally not specified |
| 18Co cells | Cells | ATCC | CRL-1459 | |
| Dulbecco's Modified Eagle's Medium (DMEM)—high glucose | Cell culture | Sigma–Aldrich | D5671 | Original brand not specified |
| Fetal Bovine Serum (FBS) | Cell culture | Sigma–Aldrich | F0392 | Original brand not specified |
| L-glutamine | Cell culture | Sigma–Aldrich | G7513 | Original brand not specified |
| Propidium iodide | Chemical | Sigma–Aldrich | P4170 | Original brand not specified |
| FACS sorter | Equipment | BD Biosciences | FACSaria | |
| 96-well ultralow-attachment plate, flat bottom | Labware | Corning | 3474 | Original catalog # not specified |
| DMSO | Chemical | Sigma–Aldrich | D8418 | Originally not specified |
| PHA-665752 | Inhibitor | Sigma–Aldrich | PZ0147 | Replaces Pfizer brand used in original study |
| HGF, human | Growth factor | Sigma–Aldrich | H5791 | Replaces Relia Tech. Inc. brand used in original study |
| T75 flask | Labware | Corning | 430641U | Originally not specified |
| Human HGF ELISA kit | Kit | Sigma–Aldrich | RAB0212 | Included for additional quality control measure |
| Plate reader capable of measuring absorbance at 450 nm | Instrument | | | Used for HGF ELISA |
| Hermes WiScan microscope | Instrument | IDEA Bio-Medical | | Originally not specified |

## Procedure

Note:

- TOP-GFP CSC clones, and untransduced spheroidal culture (no GFP) control, are generated in Protocol 1.
- CSC cultures maintained in CSC medium: modified neurobasal A medium supplemented with 1× N2 supplement, lipid mixture-1 (1 ml/500 ml medium), basic fibroblast growth factor (20 ng/ml), and epidermal growth factor (50 ng/ml) at 37°C in a humidified atmosphere at 5% $CO_2$.
- 18Co cells maintained in DMEM supplemented with 10% FBS and 1% glutamine at 37°C in a humidified atmosphere at 5% $CO_2$.
- 18Co cells will be sent for mycoplasma testing and STR profiling.
- An initial pilot experiment, performed once for each clone, will be performed to assess the clonogenic potential of the spheroid cultures. This will be performed with TOP-GFP[low], TOP-GFP[high], and total TOP-GFP gated populations left untreated (that is they will not be treated after being deposited in step 3 below). Depending on the outcome, the titration of cells might need to be adjusted before proceeding with the entire experiment as described.

1. After obtaining single cell suspensions in Protocol 1, dissociate TOP-GFP CSC clones and untransduced cultures with trypsin as described in Protocol 1 and resuspend $2 \times 10^6$ cells/ml in 500 μl of CSC medium for sorting.
2. Sort for single, propidium iodide-negative, and GFP-positive cells by FACS.
   a. Add 250 ng/ml propidium iodide solution directly to cells before analysis.

 i. Include an unstained control for gating.
 ii. Include untransduced spheroidal culture (no GFP) for gating.
 b. Gate for top 10% and bottom 10% for TOP-GFP expression.
 c. With an additional sample gate for total TOP-GFP-positive cells.
3. Deposit cells from TOP-GFP$^{low}$, TOP-GFP$^{high}$, or total TOP-GFP cell populations into 96-well ultralow-adhesion plates with 100 µl of one type of medium added to each plate. Plate at 1 cell × 24 wells, 2 cells × 16 wells, 4 cells × 8 wells, 8 cells × 8 wells, 16 cells × 8 wells, 32 cells × 8 wells, 64 cells × 8 wells, 128 cells × 8 wells, and 256 cells × 8 wells per 96-well plate. The following medium conditions are used:
 a. CSC medium.
 b. CSC medium + 500 nM PHA-665752.
 c. CSC medium + 25 ng/ml HGF.
 d. CSC medium + MFCM.
 e. CSC medium + 25 ng/ml HGF + 500 nM PHA-665752.
 f. CSC medium + MFCM + 500 nM PHA-665752.
 i. Prepare MCFM before treatment as follows:
 1. Seed 7.5 × 10$^5$ 18Co cells in 75-cm$^2$ flasks and incubate overnight.
 2. The next day wash cells twice with PBS and incubate for 24 hr with 10 ml of CSC medium without EGF and FGF-basic.
 3. The next day collect MFCM and clear by centrifugation for 5 min at 1400 RPM.
 4. Use at 1:2 dilution in CSC medium.
 5. The level of HGF present in MFCM will be determined by an ELISA following manufacturer's instructions.
4. Incubate at 37°C for ~10 days until evaluation of clonogenic potential.
 a. Replace with the appropriate culture medium condition every 4 days during incubation.
5. Count the number of cultures with spheres formed by bright field or GFP fluorescence microscopy.
 a. Exclude any contaminated cultures from analysis.

## Deliverables

- Data to be collected:
  - STR profile and result of mycoplasma testing of 18Co cells.
  - Raw data, standard curve, and concentration of HGF in MFCM.
  - All FACS plots in gating scheme (including controls), leading to final population of propidium iodide-negative, GFP-positive cells.
  - Raw counts of spheroidal cultures and total cultures examined (including if any wells were excluded).

## Confirmatory analysis plan

This replication attempt will perform the following statistical analysis listed below and compute the effects sizes for each TOP-GFP CSC clone.

- Statistical Analysis:

 Note:

1. Extreme limiting dilution analysis (ELDA) will be used to perform these tests (*Hu and Smyth, 2009*).
2. Since these tests will be performed for each of the three clones from the CSC cultures the alpha error will be adjusted with the Bonferroni correction.
 - Chi-square test for differences between any of the groups.

- Planned pairwise differences between groups with the Bonferroni correction for multiple comparisons:
  1. TOP-GFP$^{low}$ cells compared to TOP-GFP$^{hgh}$ cells.
  2. TOP-GFP$^{low}$ cells compared to TOP-GFP$^{low}$ cells with HGF.
  3. TOP-GFP$^{low}$ cells compared to TOP-GFP$^{low}$ cells with MFCM.
  4. TOP-GFP$^{low}$ cells with HGF compared to TOP-GFP$^{low}$ cells with HGF and PHA-665752.
  5. TOP-GFP$^{low}$ cells with MFCM compared to TOP-GFP$^{low}$ cells with MFCM and PHA-665752.
  6. TOP-GFP$^{whole}$ cells compared to TOP-GFP$^{whole}$ cells with PHA-665752.
- Meta-analysis of effect sizes:
  - Compute the effect sizes of the TOP-GFP CSC clones from the two newly derived cultures and the Co100 cell line (3 different clones each), compare them against the effect size in the original

paper, and use a meta-analytic approach to combine the original and replication effects, which will be presented as a forest plot.

### Known differences from the original study
All known differences are listed in the materials and reagents section above with the originally used item listed in the comments section. All differences have the same capabilities as the original and are not expected to alter the experimental design.

### Provisions for quality control
The cell line used in this experiment will undergo STR profiling to confirm its identity and will be sent for mycoplasma testing to ensure there is no contamination. Negative staining and untransduced spheroidal culture are included for gating. An initial experiment, performed once for each clone, will be performed to assess the clonogenic potential of the spheroid cultures to ensure the titration curves are appropriate. The amount of HGF in MFCM will be determined by ELISA to determine 18Co cells are producing HGF. All of the raw data, including the FACS plots, will be uploaded to the project page on the OSF (https://osf.io/pgjhx) and made publically available.

## Protocol 4: effect of MFCM on tumorigenicity in TOP-GFP$^{low}$ CSC clone
This experiment will assess the effect of MFCM on the tumorigenicity potential of one of the TOP-GFP CSC clones, which is a replication of Figure 7E.

### Sampling

- This experiment will be performed with one of the TOP-GFP CSC clones.
  - The clone used in this experiment will be from one of the two newly derived cultures with the largest difference in TOP-GFP$^{low}$ and TOP-GFP$^{high}$ as determined from Protocol 3 with untreated cells. If none of the three clones from either culture have differences similar to the reported values in the original study, then the clone with the largest difference from the Co100 cell line will be used.
- Experiment will be performed with 4 mice per injection (a total of 16 mice per cohort) for a total power of $\geq$ 88%.
  - See 'Power calculations' section for details.
- Each experiment has 3 cohorts:
  - Cohort 1: TOP-GFP$^{low}$ cells injected into nude mice.
    - 10, 100, 1000, and 5000 cells injected.
  - Cohort 2: TOP-GFP$^{high}$ cells injected into nude mice.
    - 10, 100, 1000, and 5000 cells injected.
  - Cohort 3: TOP-GFP$^{low}$ cells + MFCM injected into nude mice.
    - 10, 100, 1000, and 5000 cells injected.

### Materials and reagents

| Reagent | Type | Manufacturer | Catalog # | Comments |
|---|---|---|---|---|
| Modified neurobasal A medium | Cell culture | Life technologies | 10,888-022 | Original catalog # not specified |
| N2 supplement | Cell culture | Life technologies | 17,502-048 | Original catalog # not specified |
| Lipid mixture-1 | Cell culture | Sigma–Aldrich | L0288 | Original catalog # not specified |
| Fibroblast growth factor—Basic human | Cell culture | Sigma–Aldrich | F0291 | Original brand not specified |
| Epidermal growth factor human | Cell culture | Sigma–Aldrich | E9644 | Original brand not specified |
| Phosphate buffered saline (PBS) without MgCl$_2$ and CaCl$_2$ | Buffer | Sigma–Aldrich | D8537 | Original brand not specified |
| 0.05% trypsin/0.48 mM EDTA | Cell culture | Sigma–Aldrich | T3924 | Originally not specified |
| 18Co cells | Cells | ATCC | CRL-1459 | |
| Dulbecco's Modified Eagle's Medium (DMEM)—high glucose | Cell culture | Sigma–Aldrich | D5671 | Original brand not specified |

*Continued on next page*

*Continued*

| Reagent | Type | Manufacturer | Catalog # | Comments |
|---|---|---|---|---|
| Fetal Bovine Serum (FBS) | Cell culture | Sigma–Aldrich | F0392 | Original brand not specified |
| L-glutamine | Cell culture | Sigma–Aldrich | G7513 | Original brand not specified |
| T75 flask | Labware | Corning | 430641U | Originally not specified |
| Human HGF ELISA kit | Kit | Sigma–Aldrich | RAB0212 | Included for additional quality control measure |
| Plate reader capable of measuring absorbance at 450 nm | Instrument | | | Used for HGF ELISA |
| Propidium iodide | Chemical | Sigma–Aldrich | P4170 | Original brand not specified |
| FACS sorter | Equipment | BD Biosciences | FACSaria | |
| 96-well ultralow-attachment plate, flat bottom | Labware | Corning | 3474 | Original catalog # not specified |
| Growth factor reduced Matrigel | Cell culture | Corning | 356230 | Original brand not specified |
| 8–15 week old female athymic nude mice (20–30 grams) | Animal model | Harlan | Hsd:Athymic Nude-$Foxn1^{nu}$ | |
| 27 ½ G needle | Labware | BD Biosciences | 305109 | Original not specified |
| ACE 1 ml Luer Lock syringe | Labware | BD Biosciences | 309628 | Original not specified |

## Procedure

Note:

- TOP-GFP CSC clone is generated in Protocol 1.
- CSC cultures maintained in CSC medium: modified neurobasal A medium supplemented with 1× N2 supplement, lipid mixture-1 (1 ml/500 ml medium), basic fibroblast growth factor (20 ng/ml), and epidermal growth factor (50 ng/ml) at 37°C in a humidified atmosphere at 5% $CO_2$.
- 18Co cells maintained in DMEM supplemented with 10% FCS and 1% glutamine at 37°C in a humidified atmosphere at 5% $CO_2$.
- 18Co cells will be sent for mycoplasma testing and STR profiling.

1. After obtaining single cell suspensions in Protocol 1, dissociate TOP-GFP CSC clone and untransduced culture with trypsin as described in Protocol 1 and resuspend $2 \times 10^6$ cells/ml in 500 µl of CSC medium for sorting.
2. Sort for single, propidium iodide-negative, and GFP-positive cells by FACS.
   a. Add 250 ng/ml propidium iodide solution directly to cells before analysis.
      i. Include an unstained control for gating.
      ii. Include untransduced spheroidal culture (no GFP) for gating.
   b. Gate for top 10% and bottom 10% for TOP-GFP expression.
3. Deposit cells from TOP-GFP$^{low}$ and TOP-GFP$^{high}$ populations into 96-well ultralow-adhesion plates at 10, 100, 1000, and 5000 cells per well with 100 µl of the following medium conditions:
   a. CSC medium.
   b. MFCM.
      i. Prepare MCFM before treatment as follows:
         1. Seed $7.5 \times 10^5$ 18Co cells in 75-cm$^2$ flasks and incubate overnight.
         2. The next day wash cells twice with PBS and incubate for 24 hr with 10 ml of CSC medium without EGF and FGF-basic.
         3. The next day collect MFCM and clear by centrifugation for 5 min at 1400 RPM.
         4. Use undiluted.
         5. The level of HGF present in MFCM will be determined by an ELISA following manufacturer's instructions.
4. Incubate at 37°C for 2 hr.
   a. After this incubation period, the plates with cells and medium will be placed in a styrofoam container and transported to the facility performing mouse injection/monitoring (~30 min).
5. Afterwards mix cells and medium (100 µl) with Matrigel (100 µl) at a 1:1 ratio and inject subcutaneously into the right flank of female 8–15 week old Nude mice (20–30 grams) using a sterile 25 G needle and 1 ml syringe.
   a. Clean injection site by brief scrubbing with an isopropyl alcohol pad.

 b. 'Tent' the skin of the mouse by gentle pinching with fingers and pulling upwards.

 c. During injection, syringe is inserted into subcutaneous tissue and briefly aspirated to ensure the absence of backflow before the contents are fully injected and the needle removed from the injection site.

6. Blindly check mice weekly for a total of 9 weeks. Record if/when tumors become detectable.

 a. Monitor by palpitation and caliper measurement of depilated flanks.

 i. Caliper measurements will be used to evaluate tumor volumes from their first appearance onwards.

 ii. Calculate tumor volume as (length $\times$ width$^2$)/2.

 d. Confirm tumor presence at endpoint of study by necropsy.

## Deliverables

- Data to be collected:
  - STR profile and result of mycoplasma testing of 18Co cells.
  - Raw data, standard curve, and concentration of HGF in MFCM.
  - All FACS plots in gating scheme (including controls), leading to final population of propidium iodide-negative, GFP-positive cells.
  - Mouse health records (including when tumors become detectable and caliper measurement).

## Confirmatory analysis plan

This replication attempt will perform the following statistical analysis listed below.

- Statistical Analysis:

 Note:

1. ELDA will be used to perform these tests (*Hu and Smyth, 2009*).
   - Chi-square test for differences between any of the groups.
   - Planned pairwise differences between groups with the Bonferroni correction for multiple comparisons:
     1. TOP-GFP$^{low}$ cells compared to TOP-GFP$^{hgh}$ cells.
     2. TOP-GFP$^{low}$ cells compared to TOP-GFP$^{low}$ cells with MFCM.

- Meta-analysis of effect sizes:
  - Compute the effect sizes of each comparison, compare them against the effect size in the original paper and use a meta-analytic approach to combine the original and replication effects, which will be presented as a forest plot.

## Known differences from the original study

The experiment requires transporting the cells from a facility that is sorting the cells to another facility (~30 min away) to inject and monitor the mice. The replication will not include the TOP-GFP$^{intermediate}$ or TOP-GFP$^{low}$ with myofibroblasts that were reported for the C100.G7 clone. All known differences are listed in the materials and reagents section above with the originally used item listed in the comments section. All differences have the same capabilities as the original and are not expected to alter the experimental design.

## Provisions for quality control

The cell line used in this experiment will undergo STR profiling to confirm its identity and will be sent for mycoplasma testing to ensure there is no contamination. Additionally, cells will be tested against a rodent pathogen panel to ensure no contamination by pathogens prior to implantation into nude mice. The amount of HGF in MFCM will be determined by ELISA to determine 18Co cells are producing HGF. Negative staining and untransduced spheroidal culture are included for gating. Confirmation of tumor incidence will be confirmed at the end of the study by necropsy. All of the raw data, including the FACS plots, will be uploaded to the project page on the OSF (https://osf.io/pgjhx) and made publically available.

## Power calculations

For additional details on power calculations, please see analysis scripts and associated files on the Open Science Framework:

 https://osf.io/rfuj2/.

## Protocol 1

Not applicable.

## Protocol 2

Original data: unavailable and unable to be estimated.

### Test family

■ 2 tailed $t$ test, difference between two independent means: Bonferroni correction: alpha error = 0.003333 (corrected for the three clones from the CSC cultures and the multiple comparisons listed below).

Sensitivity Calculations performed with G*Power software, version 3.1.7 (*Faul et al., 2007*).

| Group 1 | Group 2 | Detectable effect size *d*\* | A priori power | Group 1 sample size | Group 2 sample size |
|---------|---------|------------------------------|----------------|---------------------|---------------------|
| CD133 from TOP-GFP<sup>low</sup> | CD133 from TOP-GFP<sup>high</sup> | 0.053418 | 80.0% | 10,000 | 10,000 |
| CD24 from TOP-GFP<sup>low</sup> | CD24 from TOP-GFP<sup>high</sup> | 0.053418 | 80.0% | 10,000 | 10,000 |
| CD29 from TOP-GFP<sup>low</sup> | CD29 from TOP-GFP<sup>high</sup> | 0.053418 | 80.0% | 10,000 | 10,000 |
| CD44 from TOP-GFP<sup>low</sup> | CD44 from TOP-GFP<sup>high</sup> | 0.053418 | 80.0% | 10,000 | 10,000 |
| CD166 from TOP-GFP<sup>low</sup> | CD166 from TOP-GFP<sup>high</sup> | 0.053418 | 80.0% | 10,000 | 10,000 |

\*This is the effect size that can be detected with 80% power and with a sample size of 10,000 cells analyzed per group.

## Protocol 3

Summary of original data (estimated and simulated from Figure 6D) performed with R software, version 3.1.2 (*R Development Core Team, 2014*).

• The estimated stem cell frequency and 95% lower confidence interval were used to create simulated data sets with preserved sampling structure using ELDA (*Hu and Smyth, 2009*).

| Dataset being analyzed | Total N | 95% CI<sup>lower</sup> | Estimate | 95% CI<sup>upper</sup> |
|------------------------|---------|------------------------|----------|------------------------|
| TOP-GFP<sup>low</sup> | 96 | 99.06 | 63.34 | 40.49 |
| TOP-GFP<sup>high</sup> | 96 | 2.84 | 1.94 | 1.32 |
| TOP-GFP<sup>low</sup> + HGF | 96 | 8.77 | 5.80 | 3.84 |
| TOP-GFP<sup>low</sup> + MFCM | 96 | 8.97 | 5.93 | 3.92 |
| TOP-GFP<sup>low</sup> + HGF + PHA-665752 | 96 | 155.75 | 98.15 | 61.85 |
| TOP-GFP<sup>low</sup> + MFCM + PHA-665752 | 96 | 478 | 266.21 | 148.26 |
| TOP-GFP<sup>whole</sup> | 96 | 9.14 | 6.04 | 3.99 |
| TOP-GFP<sup>whole</sup> + PHA-665752 | 96 | 8.26 | 5.47 | 3.62 |

### Test family

■ Chi-square test, differences between any of the groups: Bonferroni correction: alpha error = 0.01667 (corrected for the three clones from the CSC cultures).

Power calculations performed with R software, version 3.1.2 (*R Development Core Team, 2014*).

| Groups | $\chi^2$ test statistic | Cohen's *w* | A priori power | Total sample size |
|--------|-------------------------|-------------|----------------|-------------------|
| TOP-GFP<sup>low</sup>, TOP-GFP<sup>high</sup>, TOP-GFP<sup>low</sup> + HGF, TOP-GFP<sup>low</sup> + MFCM, TOP-GFP<sup>low</sup> + HGF + PHA-665752 TOP-GFP<sup>low</sup> + MFCM + PHA-665752, TOP-GFP<sup>whole</sup>, TOP-GFP<sup>whole</sup> + PHA-665752 | 731 | 0.975614 | 99.9% | 768 (8 groups) |

## Test family

- Chi-square test, pairwise differences between groups: Bonferroni correction: alpha error = 0.002778.

  Power calculations performed with R software, version 3.1.2 (*R Development Core Team, 2014*).

| Group 1 | Group 2 | $\chi^2$ test statistic | Cohen's w | A priori power | Group 1 sample size | Group 2 sample size |
|---|---|---|---|---|---|---|
| TOP-GFP$^{low}$ | TOP-GFP$^{high}$ | 173 | 0.949232 | 99.9% | 96 | 96 |
| TOP-GFP$^{low}$ | TOP-GFP$^{low}$ + HGF | 114 | 0.770552 | 99.9% | 96 | 96 |
| TOP-GFP$^{low}$ | TOP-GFP$^{low}$ + MFCM | 102 | 0.728869 | 99.9% | 96 | 96 |
| TOP-GFP$^{low}$ + HGF | TOP-GFP$^{low}$ + HGF + PHA-665752 | 153 | 0.892679 | 99.9% | 96 | 96 |
| TOP-GFP$^{low}$ + MFCM | TOP-GFP$^{low}$ + MFCM + PHA-665752 | 186 | 0.984251 | 99.9% | 96 | 96 |
| TOP-GFP$^{whole}$ | TOP-GFP$^{whole}$ + PHA-665752 | 0.212* | 0.251143* | 80.0%* | 96 | 96 |

*A sensitivity calculation was performed since the original data showed a non-significant effect. This is the $\chi^2$ test statistic and effect size that can be detected with 80% power.

## Protocol 4

Summary of original data (obtained from Figure 7E) performed with R software, version 3.1.2 (*R Development Core Team, 2014*).

- The estimated stem cell frequency and 95% lower confidence interval were used to create simulated data sets with preserved sampling structure using ELDA (*Hu and Smyth, 2009*).
- Both clones reported in Figure 7E were used to determine sample size to ensure an adequate number of mice are used to detect either effect size.

| Dataset being analyzed (C100.B5) | total N | 95% CI$^{lower}$ | Estimate | 95% CI$^{upper}$ |
|---|---|---|---|---|
| TOP-GFP$^{low}$ | 24 | 18,841.8 | 6939.2 | 2555.8 |
| TOP-GFP$^{high}$ | 18 | 92.2 | 37.1 | 15.1 |
| TOP-GFP$^{low}$ + MFCM | 24 | 789.3 | 310.9 | 122.6 |

## Test family

- Chi-square test, differences between any of the groups: alpha error = 0.05.

  Power calculations performed with R software, version 3.1.2 (*R Development Core Team, 2014*).

| Groups | $\chi^2$ test statistic | Cohen's w | A priori power | Total sample size |
|---|---|---|---|---|
| TOP-GFP$^{low}$, TOP-GFP$^{high}$, TOP-GFP$^{low}$ + MFCM | 75.7 | 1.070967 | 99.9% | 48 (3 groups) |

## Test family

- Chi-square test, pairwise differences between groups: Bonferroni correction: alpha error = 0.025.

  Power calculations performed with R software, version 3.1.2 (*R Development Core Team, 2014*).

| Group 1 | Group 2 | $\chi^2$ test statistic | Cohen's w | A priori power | Group 1 sample size | Group 2 sample size |
|---|---|---|---|---|---|---|
| TOP-GFP$^{low}$ | TOP-GFP$^{high}$ | 65.2 | 1.245946 | 99.9% | 16 | 16 |
| TOP-GFP$^{low}$ | TOP-GFP$^{low}$ + MFCM | 26 | 0.735980 | 97.3% | 16 | 16 |

Summary of original data (obtained from Figure 7E) performed with R software, version 3.1.2 (*R Development Core Team, 2014*).

| Dataset being analyzed (C100.G7) | total N | 95% CI$^{lower}$ | Estimate | 95% CI$^{upper}$ |
|---|---|---|---|---|
| TOP-GFP$^{low}$ | 24 | Infinity | Infinity | 12,238 |
| TOP-GFP$^{high}$ | 18 | 2499 | 961 | 370 |
| TOP-GFP$^{low}$ + MFCM | 24 | 5237 | 2352 | 1057 |

## Test family

■ Chi-square test, differences between any of the groups: alpha error = 0.05.

Power calculations performed with R software, version 3.1.2 (*R Development Core Team, 2014*).

| Groups | $\chi^2$ test statistic | Cohen's w | A priori power | Total sample size |
|---|---|---|---|---|
| TOP-GFP$^{low}$, TOP-GFP$^{high}$, TOP-GFP$^{low}$ + MFCM | 26.5 | 0.633652 | 98.19% | 48 (3 groups) |

## Test family

■ Chi-square test, pairwise differences between groups: Bonferroni correction: alpha error = 0.025.

Power calculations performed with R software, version 3.1.2 (*R Development Core Team, 2014*).

| Group 1 | Group 2 | $\chi^2$ test statistic | Cohen's w | A priori power | Group 1 sample size | Group 2 sample size |
|---|---|---|---|---|---|---|
| TOP-GFP$^{low}$ | TOP-GFP$^{high}$ | 21.3 | 0.712140 | 96.3% | 16 | 16 |
| TOP-GFP$^{low}$ | TOP-GFP$^{low}$ + MFCM | 17.5 | 0.603807 | 88.0% | 16 | 16 |

## Acknowledgements

The Reproducibility Project: Cancer Biology core team would like to thank the original authors, in particular Jan Paul Medema and Giorgio Stassi, for generously sharing critical information and reagents to ensure the fidelity and quality of this replication attempt. We thank Courtney Soderberg at the Center for Open Science for assistance with statistical analyses. We would also like to thank the following companies for generously donating reagents to the Reproducibility Project: Cancer Biology; American Type Culture Collection (ATCC), Applied Biological Materials, BioLegend, Charles River Laboratories, Corning Incorporated, DDC Medical, EMD Millipore, Harlan Laboratories, LI-COR Biosciences, Mirus Bio, Novus Biologicals, Sigma–Aldrich, and System Biosciences (SBI).

## Additional information

### Group author details

**Reproducibility Project: Cancer Biology**

Elizabeth Iorns: Science Exchange, Palo Alto, California; William Gunn: Mendeley, London, United Kingdom; Fraser Tan: Science Exchange, Palo Alto, California; Joelle Lomax: Science Exchange, Palo Alto, California; Timothy Errington: Center for Open Science, Charlottesville, Virginia

### Competing interests

JE: PhenoVista Biosciences is a Science Exchange associated laboratory. AE: PhenoVista Biosciences is a Science Exchange associated laboratory. HX: Explora BioLabs is a Science Exchange associated laboratory. NA: Explora BioLabs is a Science Exchange associated laboratory. RP:CB: EI, FT, JL: Employed by and hold shares in Science Exchange Inc. The other authors declare that no competing interests exist.

## Funding

| Funder | Author |
| --- | --- |
| Laura and John Arnold Foundation | Reproducibility Project: Cancer Biology |

The Reproducibility Project: Cancer Biology is funded by the Laura and John Arnold Foundation, provided to the Center for Open Science in collaboration with Science Exchange. The funder had no role in study design or the decision to submit the work for publication.

## Author contributions

JE, AE, HX, NA, LB, EG, Drafting or revising the article; RP:CB, Conception and design, Drafting or revising the article

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
