## [Decision Letter]

Thank you for sending your work entitled “Registered report: Wnt activity defines colon cancer stem cells and is regulated by the microenvironment” for consideration at *eLife.* Your article has been favourably evaluated by Sean Morrison (Senior Editor), Richard J Gilbertson (Reviewing Editor), and three reviewers, one of whom, Jan Paul Medema, has agreed to share his identity.

The Reviewing Editor and the reviewers discussed their comments before reaching this decision, and the Reviewing Editor has assembled the following comments that focus on the conduct of important additional experiments that will be required for further consideration of your manuscript.

Reviewer #1:

The current manuscript provides a set-up to repeat experiments performed several years ago by my team. The authors have done a careful job in listing their set-up and experimental design. Nevertheless, there are a couple of remaining concerns that need to be addressed/adjusted.

1) The name of the TOP-GFP vector is listed incorrectly. This is not a b-cat-IRES-GFP but a TOP-CMV-GFP vector.

2) In protocol 3 DMSO is added to the control. This is unlikely to have an effect, but we did not perform such an addition.

3) The titration of cells in the LD in protocol 3 is uncertain. Different spheroid cultures from individual patients have different clonogenic potential. In some case we find 1 in 4, in other cases we find 1 in 100. If the current experiments will be performed on a low clonogenic CSC (i.e. low CSC faction) the authors will need to extend their titration curves.

4) The biggest worry for the reproducibility study is the simple use of just one culture. As is clear from our study and is clear from others, there is a variability in the cultures that can be derived from colon cancers. For instance, HCT-116 and Co56 in our study do not display the same intense difference in clonogenicity and this is also reported in the study. To perform all these studies with just one culture introduces a risk into the reproducibility study. I would therefore suggest using multiple primary cultures from different patients. In addition, to facilitate the reproduction it may be wise to re-investigate the possibility of studying the clonogenic differences in one of the original cultures. This will require extensive MTAs but may facilitate the studies tremendously. I will be happy to re-invest time in checking whether this is an option. In any case, the authors should consider to use a control for clonogenic studies using AC133 staining, which has been reported by many groups to identify CSCs in spheroids, but also does not uniformaly do so (i.e. HCT116 does not work). If this fails to identify a clonogenic fraction, the TOP-GFP is also not likely to work for that particular culture.

Alternative would be to use classical cell lines to identify the top high/low difference.

5) The production of HGF by Co18 should be validated to ascertain that the TFCM is truly reproducing the original studies.

6) The MFCM stimulation of TOP-GFP^low^ cells prior to injection was performed in pure MFCM generated as described. So not admixed 1:2 with CSC medium.

7) The authors should cite other data using HGF/MET to study cancer stemness (i.e. J. Rich in Cancer Research and Stassi in Cell Stem Cell).

Reviewer #2:

This study aims to reproduce the key experiments from the article by [20] (Nature Cell Biology). Vermeulen et al. present experiments that suggest that the cancer microenviroment regulates WNT signalling, which in turn defines colon cancer stem cells.

For the reproducibility project the authors described the replication plan of key experiments. The experiments they have chosen were originally reported in Figures 2F, 6D and 7E: I agree with the choice for Figures 6D and 7E, but I feel that two other figures should be reproduced instead of Figure 2F.

In Figure 2, Vermeulen describes the generation of TopGFP transduced clones and their subsequent analysis. The analysis presented in Figure 2 goes along the following scheme:

1) Transduction of crc spheroid cells.

2) Sorting GFP^High^ and GFP^Low^ cells (Figure 2A).

3) In-vitro clonogenic assay on GFP^High^ and GFP^Low^ cells (Fig 2A, 2B and 2D).

4) Correlation of GFP^High^ with proposed cancer stem cell markers (CD133, CD24/CD29, CD44/CD166) (Figure 2F).

To me the key experiment and the basis for the study is the observation that TopGFP^High^ cells show a greater clonogenic potential than TopGFP^Low^ cells. This observation directly links WNT signaling to tumor initiating potential/cancer stem cells. Whether or not these cells express proteins that are suggested to mark cancer stem cells is not relevant as cancer stem cells are functionally defined.

By repeating Figure 2A, the in-vitro clonogenic assay, the key experiment will be reproduced. In addition to the in-vitro potential, the study should also replicate the in vivo clonogenic experiment described by Vermeulen in Figure 3A.

I suggest you repeat the following experiments:

1) In-vitro clonogenic potential (Figure 2A)

2) In-vivo clonogenic potential (Figure 3A)

3) Restore clonogenic potential of GFP^Low^ cells with MFCM and HGF in vitro (Figure 6D)

4) Restore clonogenic potential of GFP^Low^ cells with MFCM and HGF in vivo (Figure 7E)

(Results obtained from Figures 2A and 3A are controls for 6D and 7E)

In addition to this, the authors propose to generate a single spheroidal culture. Given the heterogeneity of colorectal cancer I suggest they make at least three independent cultures. Vermeulen also confirmed the results in different spheroid cultures. Including more patients will make this replication effort more effective and the results more sound.

Reviewer #3:

I am only commenting on the statistical part of this paper, Appendix A – Power Calculations.

1) For protocols 3 and 4, the authors proposed to compare means for multiple groups and do pairwise comparisons using Chi-square tests. There is no detail of the Chi-square test in the paper. As far as I know, the conventional tests for both scenarios are not a Chi-square test. The authors should provide the explicit test statistic they use as well as the assumptions of the test.

2) In Appendix A, under protocol 2, the variances of the two population distributions are not specified. Also, “the experiment will be performed with each of the 3 different TOP-GFP CSC cultures”. The multiple testing for the 3 scenarios is not addressed.

[Editors' note: further revisions were requested prior to acceptance, as described below.]

Thank you for resubmitting your article entitled “Registered report: Wnt activity defines colon cancer stem cells and is regulated by the microenvironment” for further consideration at *eLife.* Your revised Registered Report has been evaluated by Sean Morrison (Senior Editor).

Two of the three reviewers asked you to examine at least three colon cancer lines. You agreed to include one more, for a total of two. This is helpful, but at a minimum you should use three lines.

We are also disappointed that there are no in vivo tumorigenesis experiments, as indicated by Reviewer 2 (“the study should also replicate the in vivo clonogenic experiment described by Vermeulen in Figure 3A”). In vivo tumorigenesis experiments are the gold standard for drawing conclusions regarding cancer stem cells. The markers you plan to examine are controversial and of uncertain value, so no firm conclusions will be possible irrespective of what you observe with the markers. Nonetheless, if you add a third line to make the clonogenicity experiments in culture more robust, as indicated above, we would be prepared to move forward with acceptance.

---

## [Author Response]

Reviewer #1:

*The current manuscript provides a set-up to repeat experiments performed several years ago by my team. The authors have done a careful job in listing their set-up and experimental design. Nevertheless, there are a couple of remaining concerns that need to be addressed/adjusted*.

*1) The name of the TOP-GFP vector is listed incorrectly. This is not a b-cat-IRES-GFP but a TOP-CMV-GFP vector*.

We have corrected this in the revised manuscript.

*2) In protocol 3 DMSO is added to the control. This is unlikely to have an effect, but we did not perform such an addition*.

Thank you for this information. We have omitted the DMSO in the revised manuscript.

*3) The titration of cells in the LD in protocol 3 is uncertain. Different spheroid cultures from individual patients have different clonogenic potential. In some case we find 1 in 4, in other cases we find 1 in 100. If the current experiments will be performed on a low clonogenic CSC (i.e. low CSC faction) the authors will need to extend their titration curves*.

Thank you for this suggestion. We have included an additional step where each clone will be assessed to determine if the titration of cells is appropriate to detect the expected clonogenic potential. If the titration requires adjusting then this will be implemented prior to the experiment as outlined.

4) The biggest worry for the reproducibility study is the simple use of just one culture. As is clear from our study and is clear from others, there is a variability in the cultures that can be derived from colon cancers. For instance, HCT-116 and Co56 in our study do not display the same intense difference in clonogenicity and this is also reported in the study. To perform all these studies with just one culture introduces a risk into the reproducibility study. I would therefore suggest using multiple primary cultures from different patients. In addition, to facilitate the reproduction it may be wise to re-investigate the possibility of studying the clonogenic differences in one of the original cultures. This will require extensive MTAs but may facilitate the studies tremendously. I will be happy to re-invest time in checking whether this is an option. In any case, the authors should consider to use a control for clonogenic studies using AC133 staining, which has been reported by many groups to identify CSCs in spheroids, but also does not uniformaly do so (i.e. HCT116 does not work). If this fails to identify a clonogenic fraction, the TOP-GFP is also not likely to work for that particular culture.

*Alternative would be to use classical cell lines to identify the top high/low difference*.

Thank you for the suggestions. We are including the Co100 culture that was largely utilized in the original report as an additional group for the in vitro clonogenicity experiment. Also, each clone generated will be assessed for CD133 (AC133), CD29, CD24, and CD166 expression (Protocol 2), which will provide the additional assessment of clonogenic potential

*5) The production of HGF by Co18 should be validated to ascertain that the MFCM is truly reproducing the original studies*.

Thank you for this suggestion. We have included an ELISA to test for HGF levels in the MFCM that will be used.

*6) The MFCM stimulation of TOP-GFP*^*low*^
*cells prior to injection was performed in pure MFCM generated as described. So not admixed 1:2 with CSC medium*.

We have corrected this in the revised manuscript.

*7) The authors should cite other data using HGF/MET to study cancer stemness (i.e. J. Rich in Cancer Research and Stassi in Cell Stem Cell)*.

We have expanded the Introduction to include other studies.

Reviewer #2:

[…] I suggest you repeat the following experiments:

1) In-vitro clonogenic potential (Figure 2A)

2) In-vivo clonogenic potential (Figure 3A)

*3) Restore clonogenic potential of GFP*^*Low*^
*cells with MFCM and HGF in vitro (Figure 6D)*

*4) Restore clonogenic potential of GFP*^*Low*^
*cells with MFCM and HGF in vivo (Figure 7E)*

(Results obtained from Figures 2A and 3A are controls for 6D and 7E)

We agree that the clonogenic potential of TOP-GFP^high^ vs TOP-GFP^low^ cells is an important experiment, however it is depicted in figure 6D as well as Figure 2A. It includes TOP-GFP^high^ and TOP-GFP^low^ cells without HGF or MFCM treatment, thus the control is built into the experimental design and is included as a planned comparison. This is also true of Figure 7E, which includes the TOP-GFP^high^ and TOP-GFP^low^ cells without MFCM, which is included in both figures 6D and 7E. While this will require multiple tests to be performed, such as between TOP-GFP^high^ and TOP-GFP^low^ and then TOP-GFP^low^ and TOP-GFP^low^ + MFCM, this has been corrected for by adjusting the alpha error.

We also agree with the point that cancer stem cells will be functionally defined by assessing the difference in clonogenic potential between TOP-GFP^low^ and TOP-GFP^high^ cells. However, we feel the inclusion of Figure 2F allows for an additional assessment of the profile of the clones by evaluating the association with the markers originally used, which will provide another means of how this replication attempt compares to the original study.

In addition to this, the authors propose to generate a single spheroidal culture. Given the heterogeneity of colorectal cancer I suggest they make at least three independent cultures. Vermeulen also confirmed the results in different spheroid cultures. Including more patients will make this replication effort more effective and the results more sound.

Thank you for the suggestion. We are including the Co100 culture that was largely utilized in the original report as an additional group for the in vitro clonogenicity experiment. This will allow for a direct study of the clonogenic differences in one of the original cultures as well as a newly derived one.

Reviewer #3:

*I am only commenting on the statistical part of this paper,*
*Appendix A*
*– Power Calculations*.

*1) For protocols 3 and 4, the authors proposed to compare means for multiple groups and do pairwise comparisons using Chi-square tests. There is no detail of the Chi-square test in the paper. As far as I know, the conventional tests for both scenarios are not a Chi-square test. The authors should provide the explicit test statistic they use as well as the assumptions of the test*.

The reviewer is correct that there are no details of a Chi-square test in the original paper. There were no inferential statistics used. However, the software originally used for the determining the clonal frequency is the extreme limiting dilution analysis (ELDA), which is the limdil function of the statmod package in R. This is the same test that will be used in the analysis of the replication results. The test can also be seen on the website of authors of the analysis (http://bioinf.wehi.edu.au/software/elda/index.html). This approach uses the asymptotic Chi-square approximation to the log-ratio, which is applicable in these contexts as described in their paper (8). We have updated the manuscript to make this clearer.

*2) In Appendix A, under protocol 2, the variances of the two population distributions are not specified. Also,* “*the experiment will be performed with each of the 3 different TOP-GFP CSC cultures*”*. The multiple testing for the 3 scenarios is not addressed*.

For the power calculations of protocol 2, the variances of the two population distributions are unknown, thus sensitivity calculations were performed with the anticipated number of cells to be analysed with the detectable effect size reported.

We have adjusted the alpha error for testing the three clones, which is further adjusted for the multiple comparisons within each clone.

[Editors' note: further revisions were requested prior to acceptance, as described below.]

Two of the three reviewers asked you to examine at least three colon cancer lines. You agreed to include one more, for a total of two. This is helpful, but at a minimum you should use three lines.

We agree and have included the generation and analysis of an additional primary colon cancer line in the revised manuscript.

*We are also disappointed that there are no in vivo tumorigenesis experiments, as indicated by Reviewer 2 (*“*the study should also replicate the in vivo clonogenic experiment described by Vermeulen in Figure 3A*”*). In vivo tumorigenesis experiments are the gold standard for drawing conclusions regarding cancer stem cells. The markers you plan to examine are controversial and of uncertain value, so no firm conclusions will be possible irrespective of what you observe with the markers. Nonetheless, if you add a third line to make the clonogenicity experiments in culture more robust, as indicated above, we would be prepared to move forward with acceptance*.

We agree that the in vivo tumorigenesis experiements are valuable to include in this replication and protocol 4 is a direct replication of Figure 7E. This includes testing the in vivo clonogenic potential of the given clone (by comparing the TOP-GFP^low^ and TOP-GFP^high^ conditions, similar to Figure 3A), while also testing the clonogenic potential of GFP^low^ cells with MFCM (restoring the clonogenic potential of GFP^low^ cells). However, we are not including these as separate experiments (thus why it is not in reference to Figure 3A), but rather performing multiple tests (and correcting for them) to test both hypotheses.

The included markers, while controversial, are being included to allow for a comparison of the replication attempt clones to the originally reported results. While it is not feasible, and beyond the scope, for this project to examine the clonogenic potential of marker positive and negative cells, we could potentially include other markers as additional exploratory analysis and welcome any additional suggestions.